# "I Don't Feel like an Adult"—Self-Perception of Delayed Transition to Adulthood in NEET Sample

Anna Parola * , Lucia Donsì and Santa Parrello

Department of Humanities, University of Naples Federico II, 80133 Naples, Italy; donsi@unina.it (L.D.); parrello@unina.it (S.P.)
* Correspondence: anna.parola@unina.it

**Abstract:** Nowadays, there has been a debate about factors still crucial for the actual definition of adulthood and the role played by uncertainty in employment, the economic crisis, changing cultural contexts, and globalization in the adulthood transition. This study aimed to provide a deeper understanding of the self-perception of the transition to adulthood among the Italian NEET (young people not engaged in education, employment, or training). A group of 53 NEETs were asked to assess their transition from adolescence to adulthood by rating themselves on a scale from 1 (=adolescence) to 7 (=adulthood). They were also asked to reflect on the reasons why they felt like adults or not. The qualitative data were coded using the criteria of adulthood attainment proposed by Arnett's markers of adulthood scale. Analysis showed that the category with the highest frequency was role transition, and almost all the NEETs in this category felt "in-between". The excerpts portrayed the centrality of work as a crucial factor in self-perception as an adult. The research emphasizes the importance of mixed-methods research to understand where and how people feel about the transition to adulthood, aspects that are difficult to grasp using only quantitative data.

**Keywords:** transition to adulthood; NEET; role transition; mixed-method research

## 1. Introduction

The weaknesses of the current labor market (job precariousness, fixed-term contracts, income insecurity, and underemployment) appear more problematic considering that careers play an important role in youth development processes. Career development theories [1,2] suggest that the ability to implement a sound career choice is a developmental process that accelerates during adolescence and young adulthood. Starting in adolescence and emerging adulthood, the individual fulfills exploratory activities that enable them to make fitting educational and vocational choices [3]. However, career paths are no longer linear, and individuals take longer than they desire to find a job. In Western society, the school-to-work transition has become more prolonged [4]. After schooling ends, young people experience periods of high instability and uncertainty with consequences in different life domains and even on the transition to adulthood [5]. The school-to-work transition occurs at the same time as the individual is undertaking their transition to adulthood, and work turns out to be an important marker of this transition [6]. The transition to adulthood is conceived as a demographically dense period involving multiple and linked social role changes like completion of schooling, transition into the labor market, and step into family formation in a short time [7].

Over the past few decades, a growing focus has been on how young individuals transition into adulthood [6–9]. Research has shown that there are many factors associated with the delay of adulthood, including the development of independence and interdependence [10], the attainment of higher levels of psychosocial maturity [11], and the resolution of developmental tasks related to education, work, and intimate relationships [12,13].

In recent years, there has been a thoughtful reconsideration of how we define adulthood in the context of post-modernity, which invokes the dilation of the school-to-work

transition and the onset of economic independence [6,8,14]. Nowadays, the traditional markers of adulthood, such as establishing themselves in the labor market, as well as other social role transitions, such as moving away from the parental home and starting a family, are becoming increasingly elusive [5]. Indeed, in recent studies, there has been debate as to whether these factors are still essential in defining adulthood [15].

Factors such as the lack of a clear path to adulthood, job insecurity, economic downturns, shifting cultural norms, and globalization all impact the transition to adulthood [16]. As a result, the standards for reaching adulthood have become more subjective and flexible [17]. Even the perceptions of young people going through adulthood transition are different: not all adults who have achieved traditional markers feel like adults, and instead, many others who have not achieved such markers consider themselves adults [15,18]. This also occurs because there is no longer a linear transition path, but some patterns may be reversible, such as marriage or employment [19,20].

Developmental trajectories have shifted "from normative and homogeneous to more individualized and heterogeneous" [21], demanding from young people an active role in constructing transition paths [22] and the deployment of coping resources that can support these transitions [18,23]. Globalization and modernization have created a disorderly world for youth [24], and young people must try to untangle this complexity.

This is particularly true when considering southern European countries, particularly Italy and Spain, in which the late emancipation of youth is due to a complex interaction between structural and cultural factors [20,25–27]. Among the four different transition regimes proposed by Walther [28,29], Italy was included in the sub-protective transition regime characterized by "neither choice nor flexibility nor security" [29] (p. 138). The described situation is common in Mediterranean countries where young people face a lack of education and job opportunities. This often results in a prolonged dependence on their families during early adulthood, known as the "waiting phase" [30]. Livi Bacci [31] proposed an Italian pattern of delayed transition to adulthood consisting of prolongation of education, deferral of entry into the labor market, high rates of unemployment, continued co-residence with parents until the late 20s or 30s, postponing entry into a committed partnership, and delayed transition to parenthood. The extensive research conducted on the transition to adulthood in Italy confirms the difficulty of young people in making this passage, which has significant effects on well-being [21] and internalizing/externalizing health problems [32].

*Current Study*

As mentioned above, the definition of adulthood has become more fluid, and the experience of the transition to adulthood and its markers is becoming increasingly subjective. As extensive as research on transitions to adulthood appears, few studies have explored young adults' accounts of how psychological growth accompanies subjective adulthood. As Lowe and colleagues [15] argue, the process of transition to adulthood is likely to be shaped by age and life events. However, little is known about whether age variations and psychological experiences with markers and roles contribute to young people's subjective adulthood transition. Still, more studies are needed on the transition to adulthood experience of young people not engaged in education, employment, or training (NEET). Quantitative studies have shown that youth unemployment status influences the perception of the transition to adulthood to such an extent that young NEETs feel further away from adulthood than those employed [32].

In this study, we used a mixed-methods approach to explore NEET feelings of subjective adulthood. This study aimed to provide a deeper understanding of self-perception of the transition to adulthood in the Italian context by listening to the voices of NEETs. Specifically, the study focuses on Southern Italy, where the NEET rate is higher than in Northern regions. The Campania region has the highest NEET rate in the country, currently standing at 34.1%.

In line with Lowe and colleagues [15], the method was chosen to elicit various instances of adults feelings and capture traditional and psychological markers associated with subjective adult experiences.

Considering the importance that finding a job plays for the individual [33,34] and the extensive research in the Italian context that confirms this [17,27,35,36], we expect that despite age, the unemployment condition plays a crucial role in defining oneself as an adult.

## 2. Materials and Methods

### 2.1. Participants and Procedure of Data Collection

The participants were 53 Southern Italian NEETs: 20 males and 33 females, aged 18–34 (M = 27; SD = 3.22). Most of the participants are unmarried (92.5%), have no children (98.1%), and still live with their parents (77.4%). 88.5% state that their parents provide their maintenance income.

A convenient sampling method was used [37]. The participants were recruited through a network of associations in the Campania context. The administration of the survey was carried out online. Specifically, subjects were asked to take part in the research via email. Those who expressed their consent were sent a Google Forms link to participate in the survey.

Participation in the study was voluntary, and any form of compensation for participating was accepted.

Inclusion criteria were: (a) being aged between 18 and 34 y.o.; (b) being a native Italian speaker; (c) being NEET; and (d) providing informed consent. Exclusion criteria were: (a) not matching the NEET definition; (b) not completing the assessment procedure.

### 2.2. Measures

The survey primarily collected sociodemographic information like age, sex, marital status, presence of children, maintenance income, and accommodation. In addition, the following measures were administered.

### 2.2.1. Self-Perception of Adulthood

To measure how young people view their transition into adulthood, the Self-Perception of Adulthood scale [38] was used. Participants are asked to rate themselves on a scale from 1 (=adolescence) to 7 (=adulthood).

This instrument is a single-item measure that has been demonstrated to be trustworthy in the Italian context for detecting the self-perception of individuals [32,38,39].

### 2.2.2. Narrative Prompt

In an open-ended question, participants were asked to describe their positioning. Specifically, participants were asked to reflect on why they "felt like an adult" or "not yet".

The question is formulated in line with previous studies [15,17,22] that used an open-ended question to understand subjective adult identity better. No time was required to complete the measure.

### 2.3. Data Analysis

In the first step, quantitative data were coded based on the score expressed on the scale from 1–7 (from adolescence to adulthood). The data were re-coded as "I don't feel like an adult" (scores 1–2), "I feel in-between" (scores 3–5), and "I feel like an adult" (6–7).

In a second step, the qualitative data were coded using the criteria of adulthood attainment proposed by Arnett's markers of adulthood scale [40] and already used in the Italian context [17,41]. The criteria used were role transition, cognitive, emotional, behavioral, biological, legal/chronological, and responsibility [40]. According to the categorization, the "Role transition" criterion refers to transitioning into new roles. Specifically, it includes the following categories: financially independent from parents, no longer living in my parent's

household, finished with education, married, have at least one child, settle into a long-term career, employed full-time, purchased a house. The "Cognitive" criterion includes the following category: deciding on personal beliefs and values independently of parents or other influences. The "Emotional" criterion refers to having control over one's emotions, adapting the relationship with the family considering the "new" role, and establishing a committed love relationship. Specifically, it includes the following categories: establishing a relationship with parents as an equal adult; learning to always have good control of your emotions; not being deeply tied to parents emotionally; and being committed to a long-term love relationship. The "Behavioral/norm compliance" criterion involves enacting adaptive and non-risk behaviors. Specifically, it includes the following categories: avoid becoming drunk, avoid using illegal drugs, have no more than one sexual partner, drive an automobile safely and close to the speed limit, avoid drunk driving, avoid committing petty crimes like shoplifting and vandalism, avoid using profanity/vulgar language, and use contraception if you are sexually active and not trying to conceive a child. The "Biological" criterion refers to the physical growth and sexual maturity of individuals. Specifically, it includes the following categories: being capable of fathering children (men), growing to full height, and having had sexual intercourse. A "Legal/chronological criterion" refers to perceiving oneself as an adult at a certain age. Specifically, it includes the following categories: obtained a driver's license, reached age 18, reached age 21, reached age 25, and reached age 30. Finally, the "Responsibility" criterion refers to taking responsibility for one's actions and making decisions on one's own. Specifically, it includes the following categories: accept responsibility for the consequences of your action, make lifelong commitments to others; be capable of keeping a family physically safe (man); be capable of keeping a family physically safe (woman); be capable of supporting a family financially (woman); be capable of caring for children (woman); be capable of caring for children (man); be capable of running a household (woman); be capable of running a household (man).

Each narrative was coded for one or more categories. This means that, in coding each narration, it is possible to count multiple criteria that define the self-perception of adulthood. The excerpt represents the part of the text (narrative) where the criteria can be found.

The coding process involved several steps. The first author and two Ph.D. students, experts in qualitative research and the transition-to-adulthood matter, studied the Arnett [40] scoring of the adulthood criteria. Once the first author collected all the interviews, the coders analyzed them independently.

Kappa values [42] were used to calculate the agreement between coders based on the entire set [43,44]. The following benchmark was used: a lower acceptable bound of 0.70 [45]. In this study, inter-coder reliability was kappa = 0.86. The discordances between the coders were discussed and resolved.

## 3. Results

### 3.1. Self-Perception of Adulthood

For the whole sample, 73.6% (*n* = 39) reported feeling in-between, 22.6% (*n* = 12) felt like an adult, and 3.8% (*n* = 2) reported that they did not feel like an adult. The distribution of the participants in the different classes ("I don't feel like an adult" [scores 1–2], "I feel in-between" [scores 3–5], "I feel like an adult" [6,7]) is different by age but is not statistically significant. The mean of the participants who positioned themselves as not yet adults was 25.50 y.o, of those who positioned themselves in-between was 26.85, and of those who felt they were adults was 27.75 y.o.

No differences emerged between the other variables considered.

### 3.2. Narratives

The coding of the narratives is shown in Table 1. As explained above, each narrative will be coded into one or more categories based on the themes addressed in the excerpts by the participants. The category most frequently found in the excerpts was role transition,

while the biological category was not represented. Following are the results for each category, and excerpts are shown.

Role transition

Most narratives regard the role transition as crucial to feeling adult. In this marker are all those role acquisitions that individuals need to achieve their independence: being financially independent of their parents, no longer living in their parent's household, being married, having a child, settling into a long-term career, being employed full-time, and purchasing a house.

A frequent mention in the narratives of the importance of feeling like an adult when obtaining economic independence is shown. For example:

*"I consider myself mature enough to qualify as an adult. Unfortunately, the lack of economic independence, a job, and a home does not allow me to define myself as an adult... as if I were still in a transitory and not well-defined phase"* (ID 8).

*"I believe that the definitive entry into adult life can be achieved when one has stabilized, at least minimally, in the world of work, to guarantee one's economic independence, which I do not currently possess"* (ID 53).

Economic independence is linked to being unable to leave the family home, looking for a new home, and building a new family. For example:

*"I don't think I can consider myself fully adult if I don't have a job and the possibility to support myself, a necessary step to look for a house, to be able to build a family and have independence"* (ID 45).

*"I feel that I have the maturity to be an adult, but I don't have the necessary tools to be one (job, independence, etc.), so I still stay in the role of the adolescent daughter, although my desires and needs are different, i.e., home and family"* (ID 50).

Cognitive

In three narratives, the cognitive category was evident. This category refers to 'adult thinking', i.e., deciding on personal beliefs and values independently of parents or other influences. For example:

*"I live, and above all, I believe that I still do not think and reason as an adult"* (ID 13).

*"I do not know what it is like to be an adult"* (ID 2).

Emotional

Emotional refers to life experiences passed through and lived as an adult, from building a stable relationship to dealing with tough situations like losing a parent. In six excerpts, personal experiences are mentioned as experiences that have made individuals aware of feeling adult. For example:

*"I had many difficulties, for example, losing my mother, which made me grow up early"* (ID 35).

*"... life has forced me to grow up, have many responsibilities, and face situations bigger than myself, like my parents' divorce. But basically, it has served me well. I am proud of who I am today"* (ID 9).

Behavioral

Behavioral/norm compliance refers to avoiding behaviors considered 'at risk' and typical of enjoyment of life (e.g., drinking, driving under the influence, not using contraception). However, this dimension is hardly present in the excerpts:

*"Becoming an adult means leaving the carefree life behind. I should go out and drink less, be more responsible in my behaviour, but I am still young. I will think about it later..."* (ID 10).

Legal/chronological

This dimension shows a chronological justification for feeling (or not) adult and is present in most cases. It often turns out to be the beginning of a broader and more in-depth narrative of the participants, starting from a chronological justification and then moving to other categories (such as role transition). For example:

"*I am 27 years old, and chronologically I should be an adult (…)*" (ID 2).

"*Chronologically, I am no longer an adolescent, but I am still dependent on my household*" (ID 20)

Responsibility

Falling into the category of responsibility are all those excerpts that refer to adherence to responsibilities to feel adult, such as being able to protect one's family or make lifelong commitments to others. It is the second-most represented category in the narratives (17 excerpts). For example:

"*For the past year, I have become more aware of what my responsibilities are*" (ID 51).

"*I have many responsibilities related to my family and work*" (ID 33).

**Table 1.** Coding of excerpts.

| ID | Score | RT | C | E | Be | Bi | L/C | R |
|----|-------|----|---|---|----|----|-----|---|
| 1 | B | | | | | | * | |
| 2 | A | | | | | | * | |
| 3 | A | | | * | | | | |
| 4 | B | | | | | | * | |
| 5 | NA | | * | | | | | |
| 6 | B | * | | | | | | |
| 7 | B | * | | | * | | * | |
| 8 | B | * | | | | | | * |
| 9 | A | | | * | | | | |
| 10 | A | | | | * | | | * |
| 11 | B | | | | | | * | |
| 12 | A | | | * | | | * | |
| 13 | B | | * | | | | | * |
| 14 | B | * | | | | | | * |
| 15 | B | * | | | | | | * |
| 16 | B | * | | | | | | |
| 17 | B | | | | | | | * |
| 18 | B | * | | | | | | |
| 19 | B | * | | | | | | |
| 20 | B | * | | | | | * | |
| 21 | B | * | | | | | | |
| 22 | A | | | | | | | * |
| 23 | B | * | | | | | | |
| 24 | B | * | | | | | | * |
| 25 | A | | | | | | | |

**Table 1.** *Cont.*

| ID | Score | RT | C | E | Be | Bi | L/C | R |
|----|-------|----|----|----|----|----|-----|----|
| 26 | A | | | * | | | | * |
| 27 | B | * | | | | | | |
| 28 | B | | | | | | | * |
| 29 | B | * | | | | | | |
| 30 | B | * | | | | | | |
| 31 | B | | | | | | * | * |
| 32 | A | | | | | | * | |
| 33 | B | * | | | | | | * |
| 34 | B | * | | | | | | * |
| 35 | A | | | * | | | | |
| 36 | B | | | | | | | * |
| 37 | B | * | | | | | | |
| 38 | B | * | | | | | | |
| 39 | A | | | | | | * | * |
| 40 | B | * | | | | | | |
| 41 | B | * | | | | | | |
| 42 | B | * | | | | | | |
| 43 | NA | | * | | | | | |
| 44 | B | * | | | | | * | |
| 45 | B | * | | | | | | |
| 46 | B | | | | | | | * |
| 47 | B | * | | | | | | |
| 48 | B | | * | | | | | |
| 49 | B | * | | | | | | |
| 50 | B | * | | | | | | |
| 51 | A | | | | | | | * |
| 52 | B | * | | | | | | |
| 53 | B | * | | | | | | |
| **Fr.** | | 29 | 4 | 6 | 2 | - | 11 | 17 |

Note. Note. About the score: NA = "I don't feel like an adult" (scores 1–2); B = "I feel in-between" (scores 3–5); A = "I feel like an adult" (6–7). About the categorization = RT = Role transition; C = cognitive; E = emotional; Be = Behavioral; Bi = Biological; L/C = legal/chronological; R = responsibility; Fr = frequency; * refers to the codified criterion.

## 4. Discussion

The study addresses the self-perception of transition to adulthood among the so-called NEETs—individuals not engaged in education, employment, or training. Overall, although different markers and meanings are associated with adulthood, all the participants could describe what it means for them to feel like adults and through which markers they feel/do not feel like adults.

The quantitative findings reveal no differences in age for the three categories considered, i.e., "I don't feel like an adult", "I feel in-between" and "I feel like an adult". Although age seems not to play a role in defining adulthood against our hypotheses, this result should be read in light of the mean age of the respondents (M = 27; SD = 3.22). The score expressed depends more on other factors, as the same analysis of the narratives detects.

Many participants reconstruct their definition of adulthood by starting with a chronological criterion ("Although I am 30 years old...") and then explaining their reasons for choosing a score that defines them as "in-between". In most of the narratives that start from a chronological situating, there is a connection to the difficulties of defining oneself as an adult for other markers considered key in the self-perceptions, i.e., role transition.

Analysis of the narratives showed that the category with the highest frequency was role transition, i.e., the transition to adulthood conceived as the transition into new roles. Moreover, almost all excerpts coded in the role transition category feel "in-between". The excerpts showed the centrality of work as a critical factor in self-perception as an adult. The lack of a job makes leaving the family of origin difficult because they are not financially independent. Participants complain of difficulty planning their future, moving to a new home, and creating a new family because they are not financially independent. The role of work in the transition to adulthood was not considered a central marker in research in the late 1990s. Arnett [40] states that other markers, such as cognitive and responsibility, appear to be the most chosen by their study participants. Also, other studies [46] revealed that emerging adults do not generally consider role transition a crucial marker of adulthood. The centrality of role transition in the self-perception of adulthood is explained by considering the culture in which transition occurs. Several studies showed that people with collectivist or more traditional values emphasize role transition markers [47]. In addition, these results should be read and considered in light of the Italian economic conditions described above. In light of today's society and the precariousness of work and life that today's youth are forced to experience, the job takes on even more importance. Therefore, career construction, seen as a developmental task, impacts self-perception as an adult.

The second-most represented category in the narratives was responsibility. In addition, many of those who felt like adults clarified how their feeling of being an adult was related to their awareness of their responsibilities. Specifically, starting a new family and having a job made participants aware of their role because they were responsible for others (children, partners). This aligns with the findings of Arnett and Galabos [46], which suggest that some young individuals view taking on responsibilities as a positive aspect of adulthood and a way to move away from their previous youth status. As mentioned above, some excerpts have been coded as chronological/legal, referring to adulthood as the attainment of an age threshold. The reference to chronological age is evident in these excerpts. In particular, going beyond 25 is seen as the threshold at which an individual should feel like an adult, whereas it is after 30 for others. It is, however, seen as a 'should' that often does not coincide with feeling truly adult because other factors are involved.

Cognitive and emotional are the categories with the lowest number of excerpts. Particularly among the participants who use emotional criteria are those who feel like adults. The latter report that significant life experiences, especially painful ones such as the loss of a parent, led them to "grow up too fast" to the extent that they felt almost forced to grow up. The definition of adulthood is, therefore, linked to life experiences that force the transition to autonomy and independence, which are perceived to be crucial to feeling like an adult. This intraindividual change due to life circumstances documents even more the difficulty of thinking of objective criteria for transition.

Finally, cognitive markers are present in the excerpts of those who feel they are still in the adolescent phase, even though chronologically they are no longer there. The narratives of these individuals refer to confusion about adulthood and a lack of self-definition. Finally, behavior-related markers present in only two excerpts suggested difficulty in leaving the lifestyle of youth characterized by enjoyment.

The biological criterion (capable of fathering/bearing children, growing to full height, and having had sexual intercourse) identified by Arnett [40] was absent in our participants' descriptions. This result is not surprising. According to Settersten and colleagues [48], puberty today marks the transition from childhood to adolescence rather than from childhood to adulthood.

These findings add insights into barriers to accessing adulthood in the current scenario. The presence of some categories but not others well symbolizes the current scenario in which the transition occurs. External conditions, such as the barriers of the labor market, play a key role in the self-perception of transition to adulthood because it is only through economic independence that young people feel free to leave their families and move toward complete autonomy. Life experiences, like the emotional criterion, appear to be the most significant of all personal factors. They play a vital role in forming an individual's identity and encouraging autonomy during adulthood.

Critical reading can be done in light of the theoretical framework. The school-to-work transition occurs in the same period as the transition to adulthood, and they are necessarily seen as two mutually influencing developmental stages. From a developmental psychology point of view, the first work experiences help prepare for adult roles [6]. Furthermore, unemployment, low-quality jobs, and precariousness experienced by youth may constrain experiences that define one's identity and enable the attainment of adulthood, leading to a sense of failure.

Our results support the evidence that the transition to adulthood must be conceived as a complex and multifaceted developmental task [49] because it depends on the challenging, prolonged, and highly unequal spectrum of pathways that individuals take today [50]. Moreover, the transition to adulthood can be experienced very differently among individuals [51]. The centrality of financial matters is manifest in the NEET respondents. Few previous studies have considered the contribution of the financial issue to the perception of emerging adulthood characteristics [50,52]. In particular, Vosylis and Klimstra [50] showed that financial well-being promotes decreased "in-between" feelings in emerging adulthood.

Specifically, in the Italian context, our findings are in line with Di Blasi and colleagues [27] (p. 1054), who claimed that "destructuration and individualization of young people's transition paths within a worsening of the current recession might produce an increase of failed life-course trajectories due to continued difficulties and disappointments that undermine the pursuit of personal fulfilment". Unlike previous studies conducted among Italian college students, which found that many were confident in their transition to adulthood [17], our research on NEETs revealed that they struggle with assuming adult roles due to the challenges of finding employment.

## 5. Limitations, Implications, and Conclusions

Some limitations of the study must be acknowledged. First, the study was cross-sectional. Therefore, longitudinal studies were needed to support a more specific set of conclusions around the transition to adulthood of NEETs. Second, the study involved participants from Southern Italy. Our results should be replicated in other geographical areas to determine their generalizability. Third, the sample was predominantly female, which did not allow us to study gender differences. Future studies should conduct appropriate sampling to explore gender differences. Finally, this study considers a small sample of NEETs, which is a vulnerable population and does not pretend to be representative of the entire population.

Notwithstanding the previous limitations, this study shows that using both quantitative and qualitative methods is important to examine the transition to adulthood. The open-ended question allowed us to grasp individuals' self-perception, guiding the interpretation of the quantitative data obtained through positioning. This choice aligns with the ongoing debate about the transition to adulthood aimed at understanding 'what' makes an individual feel adult and how perception is subjective. Indeed, quantitative data does not fully explain what the individual needs to feel adult or, instead, what is possible with a mixed-methods approach.

Finally, these results have relevant theoretical, practical, and social implications. First, it is important to keep up research on how young people experience the transition to adulthood, including understanding the effects of delayed transitions on different psychological dimensions, such as mental health. It is also essential to study families' helpful or

harmful effects on the transition to adulthood, especially in countries with a sub-protective transition regime, such as Italy. In terms of intervention, career choices, and subsequent school-to-work transitions are the real socially recognized adult experiences [34]; thus, career guidance should consider this aspect. According to Savickas et al. [53], young people need support in becoming experts in dealing with transitions and developing resources to manage work and life transitions. The life design approach [54], a narrative-based activity aimed at supporting individuals to create personal meaning about life plans, is an optimal way to help individuals in transitions. The life design intervention is thus configured as a way to work on both levels, personal and career construction, because it fulfills two functions simultaneously, i.e., reflection and design function. It is a life design perspective to construct a self that could enable individuals to be agents of their own development in a way that enhances career and self-construction [55]. Finally, consistent with Settersten and colleagues [48], societies have yet to address individuals' difficulties transitioning to adulthood due to economic and social shifts reforming policies to improve pathways into adulthood.

In conclusion, this study aimed to understand how NEET individuals in Southern Italy perceive their transition to adulthood. Their voices were listened to in order to gain insights on this matter. Findings supported the centrality of role transitions as a crucial marker in adult self-perception. They remarked on the importance of using quantitative and qualitative methods to examine the transition to adulthood. This study has practical implications for career practitioners, who should consider the interrelated processes of both career transitions and the transition to adulthood.

**Author Contributions:** Conceptualization, A.P., L.D. and S.P.; methodology, A.P., L.D. and S.P.; formal analysis, A.P.; writing—original draft preparation, A.P.; writing—review and editing, L.D. and S.P. All authors have read and agreed to the published version of the manuscript.

**Funding:** This research received no external funding.

**Institutional Review Board Statement:** This study was performed in line with the principles of the Declaration of Helsinki. The questionnaire and methodology for this study were approved by the Human Research Ethics Committee of the University of Naples Federico II (Ethics approval number: 4/2019).

**Informed Consent Statement:** All participants in the study provided their informed consent.

**Data Availability Statement:** The data presented in this study are available upon reasonable request from the corresponding author. The data is not publicly available due to privacy reasons.

**Conflicts of Interest:** The authors declare no conflict of interest.

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
