# Peer review of "“I Don’t Feel like an Adult”—Self-Perception of Delayed Transition to Adulthood in NEET Sample"

_societies, doi:10.3390/soc13070167_

Round 1

Reviewer 1 Report

Dear authors,

The topic addressed in the article is of interest and I suggest you continue your research in this area.

Regarding the article:

1. please read carefully the rules imposed by the journal regarding the way authors are cited and the organization of bibliographic references;

2. it is not very clear to me how you established and defined the dimensions you analyze in the manuscript (cognitive, behavioral, emotional, etc.). I think a more elaborate explanation is needed.

3. in general, the chapter on research methodology must be revised and improved.

I wish you success.

Best regards, 

Author Response

Dear Reviewer 1, 

we are glad that you appreciated our work. 
Thank you for the suggestions that allowed us to refine the manuscript. Below is our point-by-point response. 

1. please read carefully the rules imposed by the journal regarding the way authors are cited and the organization of bibliographic references;

Thank you! We have reviewed the paper following the journal requirements.

2. it is not very clear to me how you established and defined the dimensions you analyze in the manuscript (cognitive, behavioral, emotional, etc.). I think a more elaborate explanation is needed.

Thank you. We have added detailed explanations of the Arnett criteria in the data analysis section, hoping now it is clear (in yellow in the text).

3. in general, the chapter on research methodology must be revised and improved.

Thank you for this suggestion. We have clarified the research methodology (in yellow in the text).

Reviewer 2 Report

Document ID: societies-2468406

This topic has potential interest to several readers. The topic itself is valuable in terms of identity and reference has been made to relevant literature. The topic holds the potential to make a valuable contribution to the existing literature.

Strengths:

The article is well structured.

The conceptual framework is clearly described.

The phases of research and the research design are clearly stated.

The results are clearly described.

Weaknesses:

1. In the “Introduction” section the authors should uniformize the terms “labor market” (line 20), labour market” (lines 49 e 80) and “job market” (line 35). 

2. The discussion is more descriptive then interpretative. The authors should consider enhancing the discussion section by incorporating more and more varied theory to analyze and interpret the findings, rather than solely presenting a descriptive account. By engaging with relevant theories and concepts in the field, it would help contextualize the results and provide a more insightful analysis of the implications and potential factors contributing to this phenomenon.

3. The conclusions are poor supported by the results presented in the article. I suggest merge the limitations and implications with the conclusion’s sections and improve the conclusions. 

Author Response

Dear Reviewer 2, 

thank you for delineating the strengths of our work. Regarding the critical aspects, thank you for your comments that allowed us to carefully re-read the manuscript. Below is the point-by-point response to the highlighted weaknesses.

Weaknesses:

  1. In the “Introduction” section the authors should uniformize the terms “labor market” (line 20), labour market” (lines 49 e 80) and “job market” (line 35). 

Thank you. We have uniformed the introduction using the term “labour market”.

  1. The discussion is more descriptive then interpretative. The authors should consider enhancing the discussion section by incorporating more and more varied theory to analyze and interpret the findings, rather than solely presenting a descriptive account. By engaging with relevant theories and concepts in the field, it would help contextualize the results and provide a more insightful analysis of the implications and potential factors contributing to this phenomenon.

Thank you for this comment that allowed us to review the discussions. The results were expanded and discussed in light of the literature on the topic. In yellow are all integrations.

  1. The conclusions are poor supported by the results presented in the article. I suggest merge the limitations and implications with the conclusion’s sections and improve the conclusions.

Thank you for this comment. We have merged the limitations and implications section with the conclusions and worked to deepen the key concepts.

Round 2

Reviewer 1 Report

Dear authors,

Congratulations for the article!

Best regards, 

Gabriela Neagu

Reviewer 2 Report

Dear author(s),

It is evident that the authors have taken the comments and suggestions into account, resulting in a significant improvement in the quality of the article.

Considering the progress made, I believe the article is ready for publication.

Best wishes.